# Vestibular Assessment in Infants with Congenital or Early Onset Sensorineural Hearing Loss: Is Neonatal Vestibular Screening Required? A Scoping Review

**DOI:** 10.3390/audiolres15020023

**Published:** 2025-02-27

**Authors:** Lauren Farquhar, Amr El Refaie

**Affiliations:** Audiology, Department of Speech and Hearing Sciences, School of Clinical Therapies, College of Medicine and Health, University College Cork, T12 AK54 Cork, Ireland

**Keywords:** congenital, infants, neonates, scoping review, screening, sensorineural hearing loss, vestibular assessment

## Abstract

Background/Objectives: Children with congenital or early onset sensorineural hearing loss (SNHL) are at a greater risk of vestibular dysfunction (VD), hypothesized to occur from the close embryological relationship between the cochlear and vestibular systems. Even with increasing focus on early detection and rehabilitation through Universal Newborn Hearing Screening (UNHS) programmes in many countries, few studies have focused on the prevalence and feasibility of vestibular assessment in infant populations. The objectives of this review are to 1. identify the prevalence of VD infants with congenital or early onset (<12 months old) SNHL, 2. identify which vestibular assessment tests/protocols are conducted on this population, 3. report sensitivity and specificity values for identified vestibular assessment tests/protocols. Methods: Studies that included infants aged 0–12 months, with congenital or early onset SNHL of any laterality, degree, or configuration, and who underwent any method of vestibular assessment were included. The review adhered to the Joanna Briggs Institute (JBI) guidance and the PRISMA-ScR extension statement. Results: A total of 18 studies were included in the review. All articles reported that infants with congenital or early onset SNHL are at a greater risk of VD, particularly those with bilateral severe–profound SNHL. The cervical vestibular-evoked myogenic potentials (cVEMP) test was the most frequently identified vestibular assessment tool for this age demographic. Conclusions: Results from the included articles coincide with results from literature assessing older paediatric populations. cVEMPs have been reported to be a feasible, sensitive, and specific screening tool in infants with congenital or early onset SNHL. The prevalence of VD in infants with congenital or early onset SNHL justify further investigation on the feasibility of establishing a pathway for vestibular assessment for all infants referred by UNHS programmes.

## 1. Introduction

In recent years, research has increasingly focused on the prevalence and assessment of vestibular dysfunction (VD) in children with hearing loss, particularly sensorineural hearing loss (SNHL) [1,2,3,4,5]. Many studies have reported an increased prevalence of VD in older paediatric populations with SNHL, with and without cochlear implantation (CI) [6,7,8,9]. However, estimates of the prevalence of VD among these populations vary widely, ranging from 20% to 85% [10]. Despite high prevalence rates of VD reported among paediatric populations with SNHL, relatively limited focus has been observed among infant populations.

Over the past decade, many studies have attempted to substantiate the direct causal mechanisms for VD in children with SNHL [11,12,13]. It is theorized that the close embryological relationship between the cochlear and vestibular systems may contribute to this comorbidity [7,14], as any underlying aetiologies causing SNHL may also cause direct disturbance to the vestibular structures [4,15]. VD in infants can vary depending on the degree and configuration of SNHL, with severe-to-profound SNHL being the most associated with significant VD [16]. This is likely due to the shared susceptibility of both the cochlear and vestibular systems to congenital, perinatal, and acquired risk factors [17].

The vestibular system plays a crucial role in the mechanism of motor function, such as gaze and neck stability and coordination of head and neck movement relative to the body [16]. Consequently, children with SNHL and unilateral or bilateral vestibulopathy often exhibit delays in motor and balance performance, including crawling and walking [17]. In infants, this manifests as delayed gross motor development resulting from reduced ‘postural control, locomotion, and gait’ [4]. Early unilateral vestibulopathy could affect motor development due to slower development of vestibular compensation in infants, especially if a progressive vestibulopathy is observed [18,19].

The comorbidity of hearing loss, VD, and subsequent motor function delays can adversely impact an infant’s ‘spatial orientation, self-concept, and joint attention’ development [4,6]. Additionally, this association may indirectly and adversely affect cognition, literacy, learning skills, and broader psychosocial development [4].

Since the establishment of Universal Newborn Hearing Screening (UNHS) programmes in many countries, increasing attention has been given to the value of early detection and rehabilitation of infants with permanent childhood hearing loss [20]. Despite the growing body of evidence regarding the association between SNHL and VD in paediatric populations, vestibular assessments are not commonplace practice for infants with confirmed SNHL and are more commonly restricted to CI candidacy assessments [21]. However, Martens et al., 2020 [16] have been exploring the prevalence of VD and the feasibility of conducting vestibular assessments on Belgian infants with congenital or early onset SNHL. VIS-Flanders has implemented a vestibular screening pathway whereby all 6-month-old infants with congenital or early onset SNHL undergo vestibular screening using cVEMPs. It was reported that a significant proportion of the demographic screened elicited abnormal vestibular responses, particularly those with severe-to-profound SNHL and/or other comorbidities such as ‘meningitis…syndromes…congenital CMV [cCMV] and cochleovestibular anomalies’ [16].

While early vestibular assessment in infants with SNHL could enable more personalized and ‘deficit-specific’ rehabilitation plans, arguments against its widespread implementation include the role of connexin 26 gene mutations that leave the vestibular system without anomaly, cost-effectiveness concerns, and the lack of large-scale studies providing substantial evidence to support its inclusion in UNHS programmes [1].

Therefore, conducting a scoping review is a crucial first step in determining if the addition of a vestibular screening component to UNHS pathways is supported by evidence.

### Objectives

Which vestibular assessment tests or protocols are undertaken in infants with congenital or early onset (<12 months old) SNHL?What are the measures of association regarding VD in infants with congenital or early onset (<12 months old) SNHL?What are the sensitivity and specificity values of the vestibular assessment tests or protocols in infants with congenital or early onset (<12 months old) SNHL?

## 2. Materials and Methods

### 2.1. Protocol and Registration

This scoping review (ScR) was conducted in accordance with the JBI Manual for Evidence Synthesis [22] and the Preferred Reporting Items for Systematic Reviews and Meta-Analyses (PRISMAScR) extension statement [23]. The scoping review was also conducted in accordance with the UCC Code of Research Conduct policy [24]. No prior registration was required for this review as it was conducted as partial fulfilment of the MSc Audiology in University College Cork.

### 2.2. Eligibility Criteria

Before commencing, clear eligibility criteria for included data sources were established by utilizing the Population, Concept, and Context (PCC) framework.

The inclusion parameters were as follows:Population: Infants with congenital or early onset (<12 months old) SNHL, with or without cochlear implantation. For this review, infants were defined as human beings aged 0–12 months of age [25]. Any data sources combining results from infants and older paediatric age groups had to report them separately.Concept: Articles that included at least one vestibular test in evaluating vestibular function in the target population. Vestibular tests in this study were defined as any test that measures the vestibulo-ocular reflex (VOR), vestibulo-collic reflex (VCR), or vestibulo-spinal reflex (VSR) in isolation.Context: Articles were required to report one or more of the following: (i) Measures of association of VD in the target population (with/without cochlear implantation) or in comparison to healthy controls or other groups of children with SNHL (with/without CI). (ii) Types of vestibular function tests carried out on the target population (with/without CI) in isolation or in comparison with other vestibular tests. (iii) Sensitivity and specificity of vestibular test(s).Other: Primary, secondary data, or grey literature published in the English language and sourced during the searching period between 1 February and 21 April 2023 and updated in February 2025 were evaluated for inclusion. Literature was required to be accessible through full text or granted access through contact with author(s) or other libraries.

The exclusion criteria were as follows:

Articles with infants with temporary or isolated conductive hearing loss. Articles conducting vestibular assessments solely on infants with (i) hearing within normal limits or (ii) infants with hearing within normal hearing limits AND medical conditions that could be risk factors for the future development of SNHL. Articles using animal or non-human models. Position or opinion papers that do not correlate or are outside the parameters of the study objectives.

### 2.3. Information Sources

The following databases were searched for relevant literature: PubMed, CINAHL, Cochrane Library, Embase, MEDLINE, Wiley, and Taylor & Francis.

### 2.4. Search

An initial preliminary search of Google Scholar was conducted to provide an overview of the size and scope of available literature on the subject using keywords (Section A.1). JBI recommends using at least two databases for initial searching while establishing a complete search string; thus, preliminary searches of PubMed and Embase were conducted [23]. It is important to note that Google Scholar was used as a primary exercise and was not utilized during comprehensive searching. All relevant papers were assessed to establish appropriate text words, keywords, MeSH, and index terms to create a comprehensive search string to be adapted for use across the databases being searched. Once two completed search strings had been established (Section A.2 and Section A.3), a comprehensive search was conducted on the stated databases. Two separate search strings were created for databases with and without a MeSH index. Comprehensive searching of all databases was conducted from 1 to 7 February 2023, updated on 21 April 2023, and updated again on 21 February 2025.

### 2.5. Selection of Sources of Evidence

All returned articles were manually logged in Microsoft Excel and assessed for inclusion eligibility as per a phased process stated below. 

Phase 1—Identification: All selected databases were search with the resulting articles arranged in alphabetical order (author, year). All duplicates were removed manually.Phase 2—Screening: All titles and abstracts of string searches across all selected databases were screened according to the eligibility criteria. In cases where no abstract was available or there was any uncertainty regarding eligibility, articles were referred directly to phase 3.Phase 3—Eligibility: Full-text screening was conducted for articles which were considered for inclusion after title and abstract screening. In any case of uncertainty regarding the inclusion of articles, a supervisor within the UCC Audiology Department was consulted.Phase 4—Inclusion: Eligible articles were documented and included in the review.

### 2.6. Data Charting Process

The objectives of the review were analysed before designing appropriate extraction tables for charting all relevant data. The data charting process was begun by separately assessing all included articles and presenting key information relating to the review objectives. Extraction tables were utilized to systematically map and collate all relevant information across all included articles. Completed extraction tables can be viewed in Section 3.

### 2.7. Data Items

Data items extracted from each article were article type, year of publication, subject source, time range of data collection, total number of subjects, age range of subjects, aetiology and diagnosis of SNHL, laterality, degree and configuration of SNHL, cochlear implantation status, vestibular assessment type, vestibular assessment protocols and normative values, sensitivity and specificity values of reported vestibular assessments and measures of association between SNHL and vestibular dysfunction. 

### 2.8. Critical Appraisal of Included Sources of Evidence

Critical appraisal of all articles included in the review was conducted where applicable using critical appraisal forms formulated by JBI [26] (Appendix B). Critical appraisal of included articles was completed to systematically assess the overall quality and risk of bias across the included articles.

### 2.9. Synthesis of Results

The handling and summarization of charted data collated all results from the included articles. As each article reported results differently, data were summarized and collated using extraction tables. Synthesis of results followed the order of the objectives stated at the start of the review.

## 3. Results

### 3.1. Selection of Sources of Evidence

The article selection process is illustrated in Figure 1. PRISMA flow diagram. A total of 205 articles were identified in the initial search. After removing 41 duplicates, the remaining 164 articles underwent title and abstract screening. During title and abstract screening, 110 articles were excluded as they did not align with the study objectives. 9 articles were sought for retrieval from other libraries or contacting the author(s). 1 article was unable to be retrieved. 54 articles were assessed for eligibility by full text, and 14 articles met the inclusion criteria. An updated search was conducted in February 2025 and a further 4 articles met the inclusion criteria. Therefore, a total of 18 articles were included in this review. 

### 3.2. Characteristics of Sources of Evidence

This review comprised 18 articles, including seven prospective cohorts (three with specified longitudinal follow-up), one observational cohort, two cross-sectionals, five case controls, two case studies, and one methodological study [4,16,27,28,29,30,31,32,33,34,35,36,37,38,39,40,41,42] (Table 1). The researchers observed a wide geographical variation between the included articles from Europe to Asia to North America and were published from 2005–2023. Most articles were of European origin [4,16,27,28,29,30,31,32,33,36,37,38,39,40,41,42] and conducted within hospital settings, some using medical registries to recruit the sample population. Sample sizes varied significantly, with recently published articles generally having larger sample sizes. Most studies reported age-matched controls, with a few comprising older children or adults as comparator groups. The age range of study participants ranged from two to twelve months. 

### 3.3. Critical Appraisal Within Sources of Evidence

JBI critical appraisal tools [26] were used to assess the methodological quality and risk of bias in all included articles. Completed critical appraisal forms are included in Appendix B. Most articles were deemed as having a low risk of bias. Three articles were deemed as having a low-moderate risk of bias and one article was deemed as having a moderate risk of bias. All risk of bias ratings are stated in Table 2.

### 3.4. Results of Individual Sources of Evidence

From the 18 included articles, two articles did not report how SNHL was assessment and diagnosed in the study subjects. Most articles reported comprehensive reports of how VD was assessed and diagnosed in the study subjects. Three articles reported corresponding sensitivity and specificity values for the vestibular assessments used. Eight articles reported details on the type of VD recorded. All but two articles reported measures of association between SNHL and VD across the study subjects. Further details are provided in ‘3.5. Synthesis of Results’. 

### 3.5. Synthesis of Results

#### 3.5.1. Characteristics of SNHL and Diagnosis

The articles primarily reported the aetiology of SNHL as ‘congenital’ and/or ‘early onset’, with some articles focusing on specific target aetiology such as Usher syndrome Type 1, congenital cytomegalovirus (cCMV), connexin 26 mutations, rubella, and autosomal recessive hereditary nonsyndromic deafness. Diagnostic approaches for SNHL varied across the articles, with several using otoscopy, tympanometry, and TEOAEs or DPOAEs. All articles reported using some form of auditory brainstem response (ABR) assessment. Click-evoked ABR (Click-ABR) was the most reported method, with three articles using automated ABR (AABR) screening. Few articles detailed the frequencies tested during ABR assessments. Most articles reported the laterality and degree of SNHL, ranging from mild to profound. Several articles reported the proportion of participants with cochlear implantation at the time of vestibular assessment, although some articles did not report on this for subjects with severe–profound SNHL (Table 3).

#### 3.5.2. Vestibular Assessments and Corresponding Sensitivity and Specificity Measures

The timing of vestibular assessments varied across articles, with the youngest infants assessed at 2.3 months and others at six to twelve months. The most common vestibular test was cervical vestibular-evoked myogenic potentials (cVEMPs) [4,16,28,29,30,31,32,33,34,35,37,38,39,40,41,42]. Five articles conducted video head impulse testing (vHIT) on infants aged six to twelve months [28,29,30,33,38]. Five articles conducted calorics on infants [38,39,40,41,42]. Modified rotatory chair assessments were conducted on infants across three included articles [28,33,36] and was determined as the least commonly used test. Further information on the vestibular assessments, protocols and normative cut-off values are reported in Table 4. Testing protocols and normative cut-off values varied across articles, with only three articles reporting both sensitivity and specificity values [29,37,39]. The cVEMP technique and protocols specified in the three articles that reported sensitivity and specificity values for cVEMPs are reported in Table 5. 

#### 3.5.3. Measures of Association Between Vestibular Dysfunction and SNHL in Infants

Most articles reported increased prevalence rates of VD in infants with congenital or early onset SNHL. Three articles reported the specific types of VD observed among the sample population [27,28,29]. The methodological study primarily reported success rates of vestibular assessment in the specified target population [37]. Full details on measures of association are reported in Table 6.

## 4. Discussion

Summary of Evidence: This scoping review was conducted to preliminarily assess the size and scope of available evidence regarding vestibular assessment on infants with congenital or early onset SNHL (with or without cochlear implantation) under twelve months of age. The primary objectives of this review were to 1. identify the prevalence of VD infants with congenital or early onset SNHL, 2. identify which vestibular assessment tests/protocols are conducted on this population, 3. report sensitivity and specificity values for identified vestibular assessment tests/protocols.

Given that paediatric vestibular assessment is rarely performed outside the realms of cochlear implant candidacy, few articles have focused on the infant population. This scarcity of literature justified the need for a scoping review to map the available evidence on this subject area. This review included mainly observational studies, as they were deemed the most appropriate in providing robust evidence for the research question.

The primary purpose of conducting vestibular assessments on infants is to differentiate between normal and abnormal vestibular function while accounting for the developmental and maturation stage of the infant. Four vestibular function tests were identified across included articles: cVEMPs, vHIT, calorics, and modified rotational chair testing. The cVEMP was the most commonly observed assessment tool in the target population across the included articles. Martens et al. (2023) [33] reported a 90% success rate for cVEMPs in six-month-olds compared to 70% for vHIT and 72.9% for the rotatory chair. This agreed with Verrecchia et al. (2019) [37], who reported cVEMPs to be highly reproducible and feasible in infants as young as 2.3 months. Zagólski O (2007) [39] reported similar results among his cohort of eighteen three-month-old infants, stating that this form of assessment was feasible in this age group, although issues relating to cooperation could impact clinicians’ ability to obtain reliable and reproducible results. These reports closely correlate with studies conducted on older children, indicating cVEMPs are both feasible and child friendly.

Other vestibular assessments, such as ‘minimized rotational chair’, were deemed easy to perform and favoured by Teschner et al., 2008 [36] due the reduced ability of young children to suppress gaze fixation. This test was reported as an altered version of the ‘Rotatory Intensity Damping Test’ that involved the child sitting on the lap of the parent while the chair rotated in a single, vigorous motion, and abruptly stopped. The child was then directly observed for nystagmus without the use of ENG [36]. Similar findings were reported in older children by Maes et al. (2014) [2]. Caloric testing in infants was reported to be more invasive and presented more limitations than cVEMPs, whereby reduced responses could be misinterpreted as bilateral vestibular hypofunction [40,42]. Other issues relating to correct irrigation due to smaller ear canal size were reported as a concern potentially leading to high levels of interindividual result variability [40,42]. Dhondt et al. (2022) [29] reported vHIT to be sensitive in detecting dysfunction in six-to-twelve-month-olds, similarly reported in older paediatric populations with more pronounced VD.

The age at which vestibular assessments were conducted varied widely across the studies. Shen et al. (2022) [34] reported that vestibular organ maturation at three months allowed for response rates equivalent to adults when using bone-conducted cVEMPs. However, Martens et al. (2023) [33] recommended conducting assessments at six months old to ensure increased cooperation and reduce confounding factors related to maturation development. However, Sheykholesami et al. (2005) [35] stated that accurate and reproducible results could be obtained from 2.3-month-old infants, although longer latencies in P13 and N23 were observed due to immature maturation.

Significant variations in sensitivity and specificity were reported across the vestibular tests, indicating that these tests may be complementary to each other rather than fixed substitutions. Several articles did not report these metrics, highlighting a gap in the literature. A previous systematic review conducted by Verbecque et al. (2017) [10] on older children (>3 y) reported cVEMP sensitivity between 48% and 100%, with a specificity range of 30% to 100%. This was similar to a previous study conducted on an adult population with SNHL that yielded a sensitivity of 91.4% and specificity of 95.8% [43]. In comparison, Zagólski O (2007) [39] reported a cVEMP sensitivity and specificity of 100% when utilizing calorics in comparison. Dhondt et al. (2022) [29] reported a vHIT of 30% sensitivity and 91% specificity in infants with congenital SNHL attributed to CMV, reported similarly by Duarte et al. (2022) [21]. Furthermore, in infants with cCMV and early onset SNHL, a sensitivity of 50% and a specificity of 91% was reported. Martens et al. (2023) [33] refrained from presenting such measures, stating that the validity of such measures could not be substantiated due to conflicts between results relating to other utilized vestibular tests.

Higher prevalence rates of VD in infants with congenital or early onset SNHL were observed across the majority of included articles [16,27,28,29,30,31,32,33,34,39,40,41,42]. The evidence included in this review reported that infants with severe–profound unilateral or bilateral SNHL were at a higher risk of VD compared to infants with mild–moderate SNHL. Often this risk was exacerbated by the presence of additional medical conditions rendering the infant more susceptible to presenting with SNHL. This finding correlates with articles conducted on older populations that reported higher prevalence of VD in those with greater degrees of SNHL. The observed high prevalence rates of VD in infants with congenital or early onset SNHL across included articles indicates that vestibular assessments potentially could be conducted on this population in an accurate, reliable, and reproducible modality. Martens et al. (2019) [4] have designed a potential pathway to implement vestibular screening seamlessly into UNHS programmes globally.

Vestibular assessment fulfils a potentially crucial role in the identification and management of opportunities for infants with SNHL who have been reported to be at a higher risk of VD [7]. Substantiating this link alongside developing the most appropriate method of assessing this population has the potential to provide valuable insights into more individualized rehabilitation processes. Through earlier detection of VD in infants with SNHL, clinicians can provide targeted intervention strategies in optimizing the developmental outcomes for this population.

Additionally, vestibular assessment can help identify particular aetiologies of sensorineural hearing loss [44]. Both hearing loss and VD are linked to specific prenatal abnormalities and genetic diseases [45]. Clinicians can acquire more diagnostic data by assessing the vestibular system, which may help identify the underlying cause of the hearing loss. This information is essential for genetic counselling, predicting developmental outcomes, and directing medical and surgical procedures.

Across the included articles, cVEMP was reported as a feasible test for infants due to high rates of successful measurements whether a response was present, absent, or inconclusive [4,16,28,29,30,31,32,33,34,35,37,38,39,40,41,42]. However, due to its limitation of which part of the system it assesses, other suitable and child-friendly assessments may be considered as potentially complementary tests to minimize any overlooked VD. Factors such as insufficient SCM muscle contraction, lack of cooperation, agitation, and protracted crying increased the risk of artifacts in cVEMP recordings [37]. Since vestibular assessments in infants can be more challenging in comparison with adults, most studies reported modifications to testing protocols. As infants cannot independently contract their SCM muscles, examiners used passive techniques, such as supporting the head during assessments or positioning the infant in a supine position [29,37,39]. Additionally, most studies favoured bone-conducted (BC) stimuli over air-conducted (AC) stimuli due to the higher prevalence of middle ear dysfunction in infants, such as middle ear effusion. In comparison to adults, infants often exhibited greater variability in response amplitudes and latencies, with some articles noting varying amplitudes and latencies in this population.

Overall, the results of this scoping review demonstrate that a higher prevalence of VD is observed in infants with congenital or early onset SNHL. The results also indicate that vestibular assessment can be both feasible and diagnostically accurate in infants with SNHL.

Directions for Future Research: The review emphasizes the need for further research and robust evidence before policy changes and implementation of targeted interventions are recommended. To progress knowledge development in this area, large-scale, prospective studies with longitudinal follow-up on cVEMPs are a priority research area. Research that refines and standardizes vestibular assessment protocols and normative cut-off values for cVEMPs is required to substantiate clearer prevalence estimates in infants with SNHL and risk factors for comorbidity, such as but not limited to Pendred syndrome, Usher syndrome, Waardenburg syndrome, and CHARGE syndrome [46]. Standardizing cVEMP protocols would lead to more defined sensitivity and specificity values. Research that compares the effect of bilateral versus unilateral vestibular dysfunction on functional motor development in infants with SNHL is also required. Cost analysis investigations are recommended to determine the cost associated with implementing an additional neonatal screening pathway in UNHS programmes.

## 5. Limitations

Although this scoping review provides promising results regarding the potential value of conducting vestibular assessment in infants with sensorineural hearing loss, the review must be viewed in light of several limitations. Firstly, the review was conducted by an independent novice researcher within a relatively short timeframe, which may have resulted in the omission of relevant articles during searching. Restricting the search to full-text articles published in the English language may have further limited the pool of eligible sources. This search yielded a small number of eligible articles for inclusion, which highlights a lack of research on the subject. Furthermore, variations in sample sizes observed across the included articles may lack generalizability. Unstandardized protocols and normative cut-off values could have introduced measurement bias, directly affecting prevalence estimation and sensitivity and specificity values in the sample populations. Variability in the signal-to-noise (SNR) ratio in infants could have impacted how present responses, absent responses, and inconclusive responses were reported in the articles. Many included articles did not report controlling for confounding factors.

## 6. Conclusions

The results of this review signal the potential value of conducting targeted vestibular assessment using cVEMPs in infants with congenital or early onset SNHL in UNHS programmes. Further prospective, large-scale research, with refined vestibular assessment protocols and normative cut-off values, is required to develop clearer prevalence estimates of VD and standardize assessments in the study demographic. As implementing such an intervention would be costly and time-consuming on already over-burdened healthcare systems, it is recommended that such targeted vestibular assessments could be conducted on infants with greater degrees of hearing loss and those with specific risk factors for comorbidity.

## Figures and Tables

**Figure 1 audiolres-15-00023-f001:**
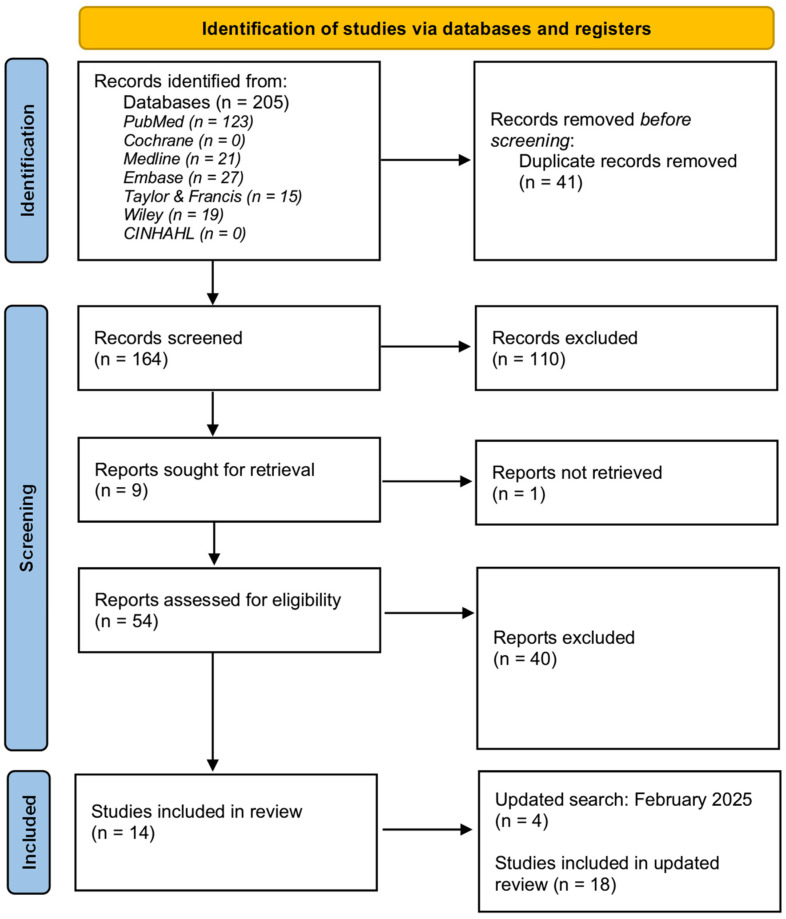
PRISMA flow diagram.

**Table 1 audiolres-15-00023-t001:** Characteristics of Included Articles.

Author, Year of Publication	Geographical Characteristics	Subject Source	Time Range of Data Collection	Study Design	Total Number of Subjects (n)	Age of Subjects/Cases
Martens et al., 2019 * [4]	Ghent, East Flanders, Belgium	UNHSP Reference Centres, Flanders	November 2017–September 2021	Prospective cohort and interventional	Approx. 400 over 4-year period	6 m
Martens et al., 2020 * [16]	Ghent, East Flanders, Belgium	UNHSP Reference Centres, Flanders	June 2018–February 2020	Prospective cohort and interventional	191	6.8 m ± 1.8 m
Akrich et al., 2023 [27]	Nantes, France	Oto-Rhino-Laryngologie pédiatrique, Chirurgie Cervico-Faciale, Necker Childrens Hospital, AP-HP, Paris, France	May 2018–February 2021	Observational cohort	22	3–10 m
Dhondt et al., 2021 [28]	Ghent, Belgium	Ghent University Hospital	June 2016 prospectively	Prospective and longitudinal	93	7 m–4.4 years
Dhondt et al., 2022 [29]	Ghent, Belgium	Ghent University Hospital	NR	Prospective and longitudinal	169	Mean: 8.9 m ± 3.27 m
Dhondt et al., 2023 [30]	Ghent, Belgium	Ghent University Hospital	June 2016–November 2021	Longitudinal cohort	185	Mean: 3.2 years, SD 1.6 years, range 0.5–6.7 years
Maes et al., 2017 [31]	Flanders, Belgium	Flemish CMV Registry	January 2007 prospectively	Cross-sectional	40 Cx26: 8 cCMV: 24 Controls: 8	Mean: 6.7 m Range: 4.8–8.9 m
Martens et al., 2022 * [32]	Ghent, East Flanders, Belgium	UNHSP Reference Centres, Flanders	June 2018–June 2021	Prospective cohort and interventional	254	7.4 m ± 2.4 m
Martens et al., 2023 * [33]	Ghent, East Flanders, Belgium	UNHSP Reference Centres, Flanders	November 2017–September 2021	Longitudinal cohort	71	6.7 m ± 1.6 m
Shen et al., 2022 [34]	Shanghai, China	Diagnosis and Treatment Centre of Hearing Impairment and Vertigo in Xinhua Hospital	May 2021– May 2022	Case control and feasibility	49 Cases: 29 Controls: 20	3 m
Sheykholesami et al., 2005 [35]	Ohio, USA	Department of Otolaryngology, Head and Neck Surgery, Case Western University Hospital of Cleveland, Ohio	NR	Prospective cohort	17 Cases: 5 Controls: 12	Mean: 2.3 m
Teschner et al., 2008 [36]	Hannover, Germany	Department of Otorhinolaryngology of the Medical University, Hannover	March 2004 –July 2005	Case study	137 Controls: 20 Cases: 117	10 m–10 years
Verrecchia et al., 2019 [37]	Sweden	UNHSP—Audiology and Neurotology Unit, Ear Nose and Throat Patient Area, Trauma and Reparative Medicine Theme, Karolinska University Hospital, Stockholm, Sweden	2015–2016 (14 months)	Feasibility	50	Mean: 2.3 ± 1.9 m Median: 2 m Range: 1–6 m Mode: 1 m
Verrecchia et al., 2020 [38]	Sweden	UNHSP—Audiology and Neurotology Unit, Ear Nose and Throat Patient Area, Trauma and Reparative Medicine Theme, Karolinska University Hospital, Stockholm, Sweden	2016–2017 (14 months)	Quantitative cross-sectional	35	4–79 m
Zagólski O, 2007 [39]	Kraków, Poland	Referred from neontologic department to a tertiary referral centre	NR	Case control	58 Cases: 18 Controls: 40	3 m
Zagolski O, 2008a [40]	Kraków, Poland	Department of Otorhinolaryngology, ‘Medicina’ Diagnostic and Therapeutic Medical Centre	NR	Case control	66 Cases: 26 Controls: 40	3 m
Zagólski O, 2008b [41]	Kraków, Poland	Department of Otorhinolaryngology, ‘Medicina’ Diagnostic and Therapeutic Medical Centre	NR	Case control	32 Cases: 17 Controls: 15	3 m
Zagolski O, 2009 [42]	Kraków, Poland	Department of Otorhinolaryngology, ‘Medicina’ Diagnostic and Therapeutic Medical Centre	NR	Case control	54 Cases: 14 Controls: 40	3 m

*: Multiple articles for same study. NR: not recorded.

**Table 2 audiolres-15-00023-t002:** Risk of Bias Judgement for Included Articles.

Author, Year	Title of Article	Bias
Martens et al., 2019 * [4]	Vestibular Infant Screening—Flanders: The implementation of a standard vestibular screening protocol for hearing-impaired children in Flanders	Low
Martens et al., 2020 * [16]	Vestibular Infant Screening (VIS)-Flanders: results after 1.5 years of vestibular screening in hearing-impaired children	Low
Akrich et al., 2023 [27]	Analysis of specific risk factors of neurodevelopmental disorder in hearing-impaired infants under ten months of age: “EnTNDre” an opening research stemming from a transdisciplinary partnership	Moderate
Dhondt et al., 2021 [28]	Vestibular Function in Children with a Congenital Cytomegalovirus Infection: 3 Years of Follow-Up	Low–moderate
Dhondt et al., 2022 [29]	Predicting Early Vestibular and Motor Function in Congenital Cytomegalovirus Infection	Low
Dhondt et al., 2023 [30]	Vestibular Follow-up Program for Congenital Cytomegalovirus Based on 6 Years of Longitudinal Data Collection	Low
Maes et al., 2017 [31]	Comparison of the Motor Performance and Vestibular Function in Infants with a Congenital Cytomegalovirus Infection or a Connexin 26 Mutation: A Preliminary Study	Low
Martens et al., 2022 * [32]	Three Years of Vestibular Infant Screening in Infants with Sensorineural Hearing Loss	Low
Martens et al., 2023 * [33]	Vestibular Infant Screening-Flanders: What is the Most Appropriate Vestibular Screening Tool in Hearing- Impaired Children?	Low
Shen et al., 2022 [34]	Cervical vestibular-evoked myogenic potentials in 3-month-old infants: Comparative characteristics and feasibility for infant vestibular screening	Low–moderate
Sheykholesami et al., 2005 [35]	Vestibular-evoked Myogenic Potentials in Infancy and Early Childhood	Low
Teschner et al., 2008 [36]	“Minimized rotational vestibular testing” as a screening procedure detecting vestibular areflexy in deaf children: screening cochlear implant candidates for Usher syndrome type I	Low
Verrecchia et al., 2019 [37]	Methodological aspects of testing vestibular-evoked myogenic potentials in infants at universal hearing screening program	Low–moderate
Verrecchia et al., 2020 [38]	The feasibility, validity and reliability of a child friendly vestibular assessment in infants and children candidates to cochlear implant	Low
Zagólski O, 2007 [39]	Vestibular system in infants with hereditary nonsyndromic deafness	Low
Zagólski O, 2008a [40]	Vestibular-evoked myogenic potentials and caloric stimulation in infants with congenital cytomegalovirus infection	Low
Zagólski O, 2008b [41]	An acoustically evoked short latency negative response in profound hearing loss infants	Low
Zagolski O, 2009 [42]	Vestibular-evoked myogenic potentials and caloric tests in infants with congenital rubella	Low

*: Multiple articles for same study.

**Table 3 audiolres-15-00023-t003:** Sensorineural Hearing Loss Characteristics in Included Articles.

Author, Year of Publication	Aetiology of SNHL (n)	Method of SNHL Diagnosis	Laterality of SNHL (n)	Degree of SNHL (n)	Cochlear Implantation
Martens et al., 2019 * [4]	Congenital	AABR (MAICO BERAphone)	NA	NA	NA
Martens et al., 2020 * [16]	Congenital and early onset < 10 m	Click-ABR, high-frequency tympanometry (1000 Hz), TE/DPOAEs. Equipment manufacturer unknown.	Unilateral: 57/169 (33.7%) Bilateral: 112/169 (66.3%)	Bilateral: mild–moderate (29%), severe–profound (37.3%) Unilateral: mild–moderate (10.6%), severe–profound (23.1%)	3
Akrich et al., 2023 [27]	Congenital, connexin 26 gene mutation, CHARGE, cCMV, Usher syndrome, GATA3 mutation, LHPL5 mutation	NR	Bilateral	Moderate–profound	Prior to cochlear implantation
Dhondt et al., 2021 [28]	cCMV and congenital SNHL: 3 (11 m, 9 m, and 10 m old)	Average click/pure tone ABR across 500, 1000, 2000, and 4000 Hz. TEOAEs, otomicroscopial inspection, and/or tympanometry. Equipment manufacturer unknown.	Unilateral: 0 Bilateral: 3	11 m infant: bilateral profound progressive 9 m infant: left profound, right moderate stable 10 m infant: bilateral profound maximal loss	11 m infant: right CI at 17 m. Left CI at 14 m. 9 m infant: left CI 10 m. 10 m infant: bilateral CI at 11 m.
Dhondt et al., 2022 [29]	cCMV	<6 m: Click-ABR during natural sleep. >6 m: Click-ABR under general anaesthetic or melatonin-induced sleep. Equipment manufacturer unknown.	Unilateral: 14 Bilateral: 10	Unilateral: 87.9 dBnHL ± 18.88 Bilateral: 79 dBnHL ± 23.78– 95 dBnHL ± 10.80.	At time of vestibular assessment, 3.6% of infants had cochlear implants (4 unilateral, 2 bilateral).
Dhondt et al., 2023 [30]	cCMV	Average click/pure tone ABR across 500, 1000, 2000, and 4000 Hz. TEOAEs, otomicroscopial inspection, and/or tympanometry. Equipment manufacturer unknown.	Unilateral: 18/30 (60%) Bilateral: 12/30 (40%)	Mild: 2/42 (4.8%) Moderate: 6/42 (14.3%) Severe: 4/42 (9.5%) Profound: 30/42 (71.4%)	20/42 (47.6%)
Maes et al., 2017 [31]	cCMV or Connexin 26 mutation	AABR (Natus ALGO)	Unilateral (cCMV): 4 Bilateral (cCMV): 4 Unilateral (Cx26): 0 Bilateral (Cx26): 8	*Mean (dBnHL)* cCMV right: 80 cCMV left: 68.8 Cx26 right: 91.3 Cx26 left: 88.1	None
Martens et al., 2022 * [32]	Congenital and early onset < 10 m	Click-ABR, high-frequency tympanometry (1000 Hz), TE/DPOAEs. Equipment manufacturer unknown.	Unilateral: 93/254 Bilateral: 161/254	Bilateral: mild– moderate (27.6%), severe–profound (35.8%) Unilateral: mild– moderate (11.8%), severe–profound (24.8%)	NR
Martens et al., 2023 * [33]	Congenital and early onset < 10 m	Click-ABR, high-frequency tympanometry (1000 Hz), TE/DPOAEs. Equipment manufacturer unknown.	Unilateral: 29/71 Bilateral: 42/71	Bilateral: mild– moderate (22.5%), severe–profound (36.6%) Unilateral: mild– moderate (12.7%), severe–profound (28.2%)	27/71
Shen et al., 2022 [34]	Congenital	Normal tympanogram-referred DPOAE (<4/6 bands), elevated air Click-ABR threshold (>30 dBnHL), AC and BC Click-ABR threshold gap within 10 dBnHL; 2000 and 4000 Hz TB-ABR, ASSR. Equipment: Interacoustics AT235H Middle Ear Analyzer, Eclipse, Interacoustics.	Unilateral: 3/17 Bilateral: 14/17	Mild–profound	NR
Sheykholesami et al., 2005 [35]	Congenital	BOA, COR, DPOAE, ABR Equipment manufacturer unknown.	NR	NR	NR
Teschner et al., 2008 [36]	Usher syndrome Type 1	BERA. Profound > 100 dB Equipment manufacturer unknown.	NR	Profound	Pre-cochlear implantation
Verrecchia et al., 2019 [37]	Congenital and early onset	Pathological response at AABR or ABR, corresponding to a HL of ≥35 dB HL for AC stimuli. Equipment: Eclipse, Interacoustics.	Unilateral: 11/50 Bilateral: 7/50	NR	NR
Verrecchia et al., 2020 [38]	Congenital, LVAS, nerve atresia, premature, Waardenburg syndrome	NR	Both unilateral and bilateral CI referrals included. No other information reported.	NR	Pre-cochlear implantation
Zagólski O, 2007 [39]	Autosomal recessive hereditary nonsyndromic deafness	Genetically diagnosed by geneticist and otoscopy, tympanometry, click-evoked otoacoustic emissions and ABR (2000–4000 Hz) Equipment: Centor-C ABR machine (Racia-Alvar, France).	Bilateral	Moderate SNHL (40–80 dBHL): 10 Severe SNHL (80 dBHL): 8	NR
Zagólski O, 2008a [40]	Congenital CMV	Otoscopy, tympanometry, click-evoked otoacoustic emissions and ABR (broadband 2000–4000 Hz clicks). Equipment: Centor-C ABR machine (Racia-Alvar, France).	Bilateral	Profound	NR
Zagólski O, 2008b [41]	Congenital	Otoscopy, tympanometry, ABR/ASNR broadband 2 KHz and 4 KHz clicks, and 500 Hz 80–110 SPL tone-burst during sleep. Equipment: Centor-C ABR machine (Racia-Alvar, France).	Bilateral	Profound	NR
Zagolski O, 2009 [42]	Congenital rubella	Otoscopy, click-evoked OAEs, ABR (BC, broadband clicks compared with AC ABR). Equipment: Centor-C ABR machine (Racia-Alvar, France).	NR	20–80 dBHL: 3 >80 dBHL: 6	NR

* Multiple articles for same study. NA: not applicable, NR: not reported. AABR: automated auditory brainstem response, ABR: auditory brainstem response, AC/BC: air conduction/bone conduction, ASSR: Auditory Steady State Response, BERA: brainstem-evoked response audiometry, BOA: Behavioural Observation Audiometry, CI: cochlear implant[ation], Click-ABR: click-evoked ABR, CMV: cytomegalovirus, COR: Conditioned Oriented Responses, dBHL: Decibels Hearing Level, HL: hearing loss, OAEs: otoacoustic emissions, SNHL: sensorineural hearing loss, TB-ABR: tone-burst ABR, TE/DPOAEs: Transient-evoked/Distortion Product Otoacoustic Emissions.

**Table 4 audiolres-15-00023-t004:** Summary of Results of Included Articles Relating to Review Objectives.

Author, Year of Publication	Age of Vestibular Assessment	Vestibular Reflex	Vestibular Tests and Organ (Objective 1)	Vestibular Protocol (Objective 1)	Normative Cut-off Values	Quality of Vestibular Tests (Objective 3)
Martens et al., 2019 [4]	6 m	VCR/VSR	cVEMPs: saccule and inferior portion of vestibular nerve	Stimulus: BC 500 Hz TB (1-2-1 ms) at 59 dBHL, 5 Hz repetition rate	Abnormal: absent cVEMP response	NA
Martens et al., 2020 [16]	6 m	VCR/VSR	cVEMPs: saccule and inferior portion of vestibular nerve	Stimulus: BC 500 Hz TB (1-2-1 ms) at 59 dBHL, 5 Hz repetition rate	Abnormal: Inconclusive OR Abnormal (<1.3 rectified wave amplitude) OR Absent	Sensitivity: NR Specificity: NR
Akrich et al., 2023 [27]	<10 m	NR	NR	NR	NR	NR
Dhondt et al., 2021 [28]	6 m and 12 m	VOR	vHIT: lateral SCC	Amplitude: 10–20 degrees Peak velocity: 150’/s	Abnormal: below 0.4	Sensitivity: 87% Specificity: NR
		VOR	Rotary Test: mid-frequency lateral SCC	Stimulus: sinusoidal harmonic acceleration at 0.01, 0,04, and 0.16 Hz. Peak velocity: 60’/s	Abnormal: below 9, 19, and 22% at specified frequencies	Sensitivity: NR Specificity: NR
		VCR/VSR	cVEMP: saccule and inferior portion of vestibular nerve	Stimulus: BC linear 500 Hz tone bursts (1-2-1 ms) at 59 dBHL, 5 Hz repetition rate Amplification: 5000 times Filter: 10–1500 Hz bandpass	Abnormal: absent cVEMP response	Sensitivity: NR Specificity: NR
Dhondt et al., 2022 [29]	8.9 m ± 3.27 m	VOR	vHIT: lateral SCC (high frequency)	Amplitude: 10–20 degrees Peak velocity: 150’/s	Mild dysfunction: 0.4–0.7 Severe dysfunction: below 0.4	Vestibular assessment and congenital SNHL: Sensitivity: 35% Specificity: 91% Vestibular assessment and early acquired SNHL: Sensitivity: 50% Specificity: 91%
		VCR/VSR	cVEMP: saccule and inferior portion of vestibular nerve	Stimulus: BC linear 500 Hz tone-bursts (1-2-1 ms) at 59 dBHL, 5 Hz repetition rate Amplification: 5000 times Filter: 10–1500 Hz bandpass	Mild dysfunction: interpeak amplitude: below 0.3 (Bio-Logic) or 1.3 (Neuro-Audio) Severe dysfunction: no reproducible response	
Dhondt et al., 2023 [30]	6–12 m	VOR	vHIT: lateral SCC (high frequency)	Amplitude: 10–20 degrees. Peak velocity: 150’/s	Mild dysfunction: 0.4–0.7 Severe dysfunction: below 0.4	Sensitivity: NR Specificity: NR
		VCR/VSR	cVEMP: saccule and inferior portion of vestibular nerve	Stimulus: BC linear 500 Hz tone-bursts (1-2-1 ms) at 59 dBHL, 5 Hz repetition rate Amplification: 5000 times Filter: 10–1500 Hz bandpass	Mild dysfunction: interpeak amplitude: below 0.3 (Bio-Logic) or 1.3 (Neuro-Audio) Severe dysfunction: no reproducible response	Sensitivity: NR Specificity: NR
Maes et al., 2017 [31]	4.8–8.9 m	VCR/VSR	cVEMPs: saccule and inferior portion of vestibular nerve	Stimulus: BC linear 500 Hz TB (1-2-1 ms) at 59 dBHL, 5 Hz repetition rate Amplification: 5000 times Filter: 10–1500 Hz bandpass	Abnormal: absent cVEMP response	Sensitivity: NR Specificity: NR
Martens et al., 2022 [32]	6 m	VCR/VSR	cVEMPs: saccule and inferior portion of vestibular nerve	Stimulus: BC 500 Hz TB (1-2-1 ms) at 59 dBHL, 5 Hz repetition rate	Abnormal: Inconclusive OR Abnormal (<1.3 rectified wave amplitude) OR Absent	Sensitivity: NR Specificity: NR
Martens et al., 2023 [33]	6 m	VCR/VSR	cVEMPs: saccule and inferior portion of vestibular nerve	Stimulus: BC 500 Hz TB (1-2-1 ms) at 59 dBHL, 5 Hz repetition rate	Abnormal: Inconclusive OR Abnormal (<1.3 rectified wave amplitude) OR Absent	Sensitivity: NR Specificity: NR
	6–9 m	VOR	vHIT: horizontal SCC (high frequency)	Amplitude: 10–20’ Peak velocity: 150–250’/s (horizontal plane) and 100–200’/s (vertical plane)	Abnormal: VOR gain <0.7. Borderline results 0.6–0.7 considered abnormal if replicated during prospective repeat test	Sensitivity: NR Specificity: NR
	NR (young children)	VOR	Rotary Test: mid-frequency lateral SCC	Stimulus: sinusoidal harmonic acceleration at 0.01, 0,04, and 0.16 Hz Peak velocity: 60’/s	Abnormal: below 9, 19, and 22% at specified frequencies	Sensitivity: NR Specificity: NR
Shen et al., 2022 [34]	3 m	VCR/VSR	cVEMPs: saccule and inferior portion of vestibular nerve	Stimulus: TB-500 Hz (rise/fall 1 ms, plateau time 2 ms) at 132 dBpeSPL (105 dBnHL), repetition rate 5 Hz. BC conductor at 60 dBnHL, 5.1 Hz stimulus rate Minimum sweeps: 50 (twice) Amplified and bandpass: 10–3000 Hz Window: −20–80 ms	Normal: the mean + 2 SD of each parameter in normal hearing infants defined as the upper-normal limit Absent: response or value exceeding the normal range was considered as abnormal	Sensitivity: NR Specificity: NR
Sheykholesami et al., 2005 [35]	2.3 m	VCR/VSR	cVEMPs: saccule and inferior portion of vestibular nerve	Stimulus: AC and BC short TB (500 Hz at 95 dBnHL) Rise/fall time: 1 ms Plateau: 2 ms Amplified and bandpass: 20 Hz to 2 KHz Window: 100 ms	Abnormal: absent cVEMP response	Sensitivity: NR Specificity: NR
Teschner et al., 2008 [36]	0–12 m	VOR	Minimized Rotational Vestibular Testing: mid- frequency SCC	Chair acceleration: 16’/s for 8 s. Chair constant rotational velocity: 150’/s for 20 s VNG/ENG not used	Abnormal: no post-rotational nystagmus observed	Sensitivity: NR Specificity: NR
Verrecchia et al., 2019 [37]	2.3 ± 1.9 months (median: 2; range: 1–6; mode: 1).	VCR/VSR	cVEMPs: saccule and inferior portion of vestibular nerve	Stimulus: BC 500 Hz TB at 50 dBnHL (119 dBFL), 2 ms rise/plateau/fall. Repetition rate: 5.1/s Amplification: 2000 gain Filter: bandpass (10–750 Hz) Window: −20–80 ms Sweeps: 120	Normal: positive–negative EMG deflection with a latency of 12–17 ms for the first peak (p1) and 20–25 ms for the second peak (n1) after stimulus Abnormal: absent repeatable responses	Amplitude cut-off: Sensitivity: 87% Specificity: 89% Scaled amplitude cut-off: Sensitivity: 94% Specificity: 96%
Verrecchia et al., 2020 [38]	3–84 m	VOR	HIT: horizontal SCC (high frequency)	Child sitting on parent’s lap and head rotated 20–30 degrees to either side while sticker is on examiner’s nose to promote gaze fixation	Directly observing nystagmus	Sensitivity: NR Specificity: NR
		VOR	vHIT: horizontal SCC (high frequency)	Turns child’s head to either side in jerk motions at 2–30 degrees. Three–five trials per side	Gain values <0.75 were related to vestibular dysfunction	Sensitivity: NR Specificity: NR
		VCR/VSR	cVEMPs: saccule and inferior portion of vestibular nerve	Stimulus: BC 500 Hz TB at 50 dBnHL (119 dBFL), 2 ms rise/plateau/fall. Repetition rate: 5.1/s Amplification: 2000 gain Filter: bandpass (10–750 Hz) Window: −20–80 ms Sweeps: 120	Normal: positive–negative EMG deflection with a latency of 12–17 ms for the first peak (p1) and 20–25 ms for the second peak (n1) after stimulus Abnormal: absent repeatable responses	Sensitivity: NR Specificity: NR
		VOR	Mini Ice Water Calorics: lateral SCC	6–10 °C water in canal for 10 s while child is lying down and is then raised to supine position and eye movements observe twice in dark using VOS mask for 5–10 s, two repetitions	Normal: 3 consecutive caloric nystagmus beats observed by VOS under 30–90 s after irrigation and reproduced once more after interval of visual fixation Abnormal: absent caloric nystagmus reproducible on two test repetitions	Sensitivity: NR Specificity: NR
Zagólski O, 2007 [39]	3 m	VOR	Calorics: lateral SCC	Stimulus: 20’C cold water irrigation (20 mL) for 20 s	Both sides compared for weakness. Nystagmus observed directly. Normal: Nystagmus latency: 5–15 s. Duration: 60–70 s. The frequency of bytes: 2–3/s. Abnormal: latency and/or duration differed from normal range by >30%	Sensitivity: 83.3% Specificity: 66.6% Likelihood ratio: 83.3%
	3 m	VCR/VSR	cVEMPs: saccule and inferior portion of vestibular nerve	Stimulus: AC TB presented at 500 Hz (frequency) at 90 dBSPL in ipsilateral ear, no masking in contralateral ear Repetition rate: 5 Hz (1 ms rise/fall rate), plateau time (2 ms)	Abnormal: absent cVEMP response	Sensitivity: 100% Specificity: 100%
Zagólski O, 2008a [40]	3 m	VOR	Calorics: lateral SCC	Stimulus: 20’C cold water irrigation (20 mL) for 20 s	Both sides compared for weakness. Nystagmus observed directly. Normal: Nystagmus latency: 5–15 s. Duration: 60–70 s. The frequency of bytes: 2–3/s. Abnormal: Latency and/or duration differed from normal range by >30%	Sensitivity: NR Specificity: NR
	3 m	VCR/VSR	cVEMPs: saccule and inferior portion of vestibular nerve	Stimulus: air conduction TB presented at 500 Hz at 90 dBSPL in ipsilateral ear, no masking in contralateral ear Repetition rate: 5 Hz (1 ms rise/fall rate), plateau time (2 ms)	Abnormal: absent cVEMP response	Sensitivity: NR Specificity: NR
Zagólski O, 2008b [41]	3 m	VOR	Calorics: lateral SCC	Stimulus: 20’C cold water irrigation (20 mL) for 20 s	Both sides compared for weakness. Nystagmus observed directly Normal: 5–15 s latency and 60–70 s duration. Frequency of beats from 2–3/s Abnormal: differed from normal response more than 30% for time response or number of beats	Sensitivity: NR Specificity: NR
		VCR/VSR	cVEMPs: saccule and inferior portion of vestibular nerve	Stimulus: 75 responses of air conduction TB presented at 500 Hz at 90 dBSPL in ipsilateral ear, no masking in contralateral ear. Repetition rate: 5 Hz (1 ms rise/fall rate), plateau time (2 ms)	Abnormal: absent cVEMP response	Sensitivity: NR Specificity: NR
Zagólski O, 2009 [42]	3 m	VOR	Calorics: lateral SCC	Stimulus: 20’C cold water irrigation (20 mL) for 20 s	Both sides compared for weakness. Nystagmus observed directly. Normal: Nystagmus latency: 5–15 s. Duration: 60–70 s. The frequency of bytes: 2–3/s. Abnormal: latency and/or duration differed from normal range by >30%	Sensitivity: NR Specificity: NR
	3 m	VCR/VSR	cVEMPs: saccule and inferior portion of vestibular nerve	Stimulus: AC TB presented at 500 Hz (frequency) at 90 dBSPL in ipsilateral ear, no masking in contralateral ear Repetition rate: 5 Hz (1 ms rise/fall rate), plateau time (2 ms)	Abnormal: absent cVEMP response	Sensitivity: NR Specificity: NR

NA: not applicable, NR: not reported, AC: air conduction/conducting, BC: bone conduction/conducted, cVEMP: cervical vestibular-evoked myogenic potential, dBHL: Decibels Hearing Level, dBnHL: Decibels Normal Hearing Level, dBpeSPL: Decibels Peak Equivalent Sound Pressure Level, dBSPL: Decibels Sound Pressure Level, ms: milliseconds, m: months, mL: millilitres, SCC: semicircular canal, SNHL: sensorineural hearing loss, TB: tone-burst, VCR: vestibulo-collic reflex, VNG/ENG: Videonystagmography/Electronystagmography, VOR: vestibulo-ocular reflex, VSR: vestibulo-spinal reflex.

**Table 5 audiolres-15-00023-t005:** Summary of cVEMP Protocols in Articles that Reported Sensitivity and Specificity Values.

	Test Specifications	Recording	Technique and Infant Position
Dhondt et al., 2022 [29]	Equipment: Bio-Logic Navigator-Pro platform, Mundelein, IL, USA and Neuro-Audio version 2010, Neurosoft, Ivanovo, Russia. EMG delivery algorithm: NR Bone-conducted stimuli: linear 500 Hz tone-burst (1-2-1 ms) at 59 dBHL (129 dBFL) and 5 Hz stimulus repetition rate.	Electrode array: noninverting electrode placed at midpoint of SCM muscle, inverting electrode 1–2 cm below sternoclavicular junction, ground electrode on forehead. Signal processing: amplification (gain: 5000), bandpass filter (10–1500 Hz). Sweeps: NR	Infant position: supine position on a sloping pillow and stimulated to rotate head sideways using video screen or toys. Technique: NR
Verrecchia et al., 2019 [37]	Equipment: Radioear B71 bone vibrator device. EMG-driven stimulus delivery algorithm: EMG reference level of 50–150 µVolts and sampled every 100 ms. Once EMG was within this range, new RMS averaging performed over 100 ms before stimulus delivery. EMG recording continued for up to 80 ms post stimulation. VEMP recording window: −20 ms–+80 ms. Process continually repeated for each sweep. Bone-conducted stimuli: tone-burst stimuli presented on the mastoid bone at 500 Hz and 50 dBnHL (119 dBFL), 2 ms rise-plateau-fall configuration, stimulation rate of 5.1/s.	Electrode array: Recordings performed unilaterally. Two inverting electrodes added to belly of two SCMs: non-inverting electrode on manubrium sterni, ground electrode on forehead. Skin prepared with gentle abrasion, maintaining electrode impedance under 10 kΩ. Signal processing: amplification (gain: 2000), bandpass filter (10–750 Hz) within −20–+80 ms recording window. Sweeps: ≥120 sweeps, max. 200 per trial, typically 90–120 s per side.	Infant position: Supine in parent’s arms, head supported on examiner’s hand, awake enough to generate neck muscle activity. Examiner modulated infant’s head support to change SCM activity. Technique: bone conduction transducer held to mastoid region by examiner with lateromedial digital pressure, placed above imaginary antero-posterior line crossing ear canal.
Zagólski O, 2007 [39]	Equipment: Centor-C ABR machine (Racia-Alvar, France). EMG delivery algorithm: NR Air-conducted tone-burst stimuli: averaged from 75 responses at 500 Hz at 90 dBSPL, no contralateral masking noise, repetition rate of 5 Hz and rise-and-fall time of 1 ms and plateau time of 2 ms.	Electrode array: Reference electrode placed over upper sternum. Two electrodes placed on symmetric sites on the upper half of both isometrically contracted SCM. Signal processing: NR Sweeps: 75	Infant position: supine position with muscle contraction obtained by bending the neck slightly backwards and holding in position. Technique: NR

EMG: electromyography, ms: milliseconds, NR: not reported, SCM: sternocleidomastoid.

**Table 6 audiolres-15-00023-t006:** Summary of Association between Infant SNHL and Vestibular Dysfunction in Included Articles.

Author, Year of Publication	Measures of Association (Objective 2)	Vestibular Dysfunction Recorded
Martens et al., 2019 [4]	NA	NA
Martens et al., 2020 [16]	Absent cVEMPs in unilateral/bilateral severe–profound compared to unilateral/bilateral mild–moderate: (*p*-Value: 0.001) Severe–profound: 32/154 (20.8%) Mild–moderate: 3/100 (3.0%)	Unilateral mild–moderate: 1/30 (3.3%) Unilateral severe–profound: 14/63 (22.2%) Bilateral mild–moderate: 2/70 (2.9%) Bilateral severe–profound: 18/91 (19.8%)
Akrich et al., 2023 [27]	9/22 (40.9%) vestibular dysfunction observed	Hyporeflexia: 3/22 (13.63%) Areflexia: 6/22 (27.27%)
Dhondt et al., 2021 [28]	Congenital SNHL and vestibular dysfunction Prevalence: 3/3 (100%)	11 m infant: right SCC and right saccule dysfunction. 9 m infant: bilateral SCC and saccule dysfunction. 10 m infant: bilateral SCC and saccule dysfunction.
Dhondt et al., 2022 [29]	Congenital SNHL and vestibular dysfunction Prevalence: 35% OR: 5.63 *p*-Value = 0.002 95% CI: 1.91–16.60 Early acquired SNHL and vestibular dysfunction Prevalence: 50% OR: 21.91 *p*-Value = 0.003 95% CI (2.88–166.46)	Laterality Unilateral: 7.7% Bilateral: 4.1% Type Lateral SCC and saccule: 6.2% Lateral SCC only: 1.8% Saccule only: 0%
Dhondt et al., 2023 [30]	Prevalence: 48.4%	NR
Maes et al., 2017 [31]	cCMV, congenital SNHL, and vestibular dysfunction (*n* = Absent cVEMP) Right: 4/7 (57.1%) Left: 3/7 (42.9%) Cx26, congenital SNHL, and vestibular dysfunction Right: 0/8 Left: 0/8	cCMV, congenital SNHL and vestibular dysfunction Bilateral SNHL (4): 2/4 bilateral VD, 1/4 present, 1/4 not performed. Unilateral SNHL (3): 1/3 unilateral VD, 1/3 bilateral VD, 1/3 present response.
Martens et al., 2022 [32]	Abnormal cVEMPs in unilateral/bilateral severe–profound compared to unilateral/bilateral mild–moderate: RR (9.8) (*p*-Value: 0.003) Severe–profound: 15/102 (14.7%), 95% CI (9.1–22.9) Mild–moderate: 1/67 (1.5%), 95% CI (0.3–8.0)	Unilateral absent cVEMP responses reported on same side as all subjects with unilateral SNHL (*n* = 6). Bilateral cVEMP configuration varied in subjects with bilateral SNHL ((unilateral abnormal (*n* = 2), unilateral absent (*n* = 2), bilateral abnormal (*n* = 1) and bilateral absent (*n* = 2)).
Martens et al., 2023 [33]	Abnormal cVEMPs Unilateral: 5/71 (7%) Bilateral: 7/71 (9.9%) Abnormal vHIT: Unilateral: 6/65 (9.2%) Bilateral: 6/65 (9.2%) Infants with abnormal cVEMP responses, vHIT results were significantly more abnormal (90% (9/10), 95% CI (60–98.2%)) than normal (10% (1/10), 95% CI (1.8–40.4%)). Rotary Test: only feasibility/success calculations made as assessment was more challenging to conduct on infants.	Unilateral mild–moderate: 9/65 Unilateral severe–profound: 17/65 Bilateral mild–moderate: 14/65 Bilateral severe–profound: 25/65
Shen et al., 2022 [34]	Response rate (AC): 62% (SNHL), 88.89% (hearing infants). Response rate (BC): 86.36% (SNHL). No significant difference to hearing infants. IAR ranges of AC and BC in SNHL in upper-normal limit of hearing infants = bilateral vestibular dysfunction is symmetrical in SNHL infants.	NR
Sheykholesami et al., 2005 [35]	1/5 subjects with SNHL had absent cVEMPs bilaterally (20%).	NR
Teschner et al., 2008 [36]	Success rates in 0–12 m: 25/29 successfully (86%). No prevalence rates age segregated.	NR
Verrecchia et al., 2019 [37]	NR (methodological aspect investigation)	NR
Verrecchia et al., 2020 [38]	NR (feasibility and reliability investigation)	NR
Zagólski O, 2007 [39]	Absent cVEMPs recorded bilaterally in 12/18 infants with SNHL. No caloric responses recorded bilaterally in 6/18 infants with SNHL. Degree of semicircular canal impairment was higher in subjects with profound SNHL.	NR
Zagólski O, 2008a [40]	Absent cVEMPs recorded bilaterally in 8/8 infants with profound SNHL: 15.4% of overall cohort. Absent calorics recorded bilaterally in 8/8 infants with profound SNHL: 15.4% of overall cohort.	NR
Zagólski O, 2008b [41]	Absent cVEMPs recorded in 22 ears (22/34, 64.7%). Abnormal/absent ASNR recorded in 24 ears (24/34, 70.58%). Absent calorics recorded in 26 ears (26/34, 76.47%).	NR
Zagólski O, 2009 [42]	Absent cVEMPs reported in 100% of infants with severe–profound SNHL (12/12): 43% of overall cohort. Absent calorics reported in 100% of infants with severe–profound SNHL (12/12): 43% of overall cohort. Statistically signification association between degree of SNHL and vestibular dysfunction (*p* < 0.001, r = 0.9).	NR

NA: not applicable, NR: not reported.

## Data Availability

The raw data supporting the conclusions of this article will be made available by the authors on request.

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
