# Peer review of "Vestibular Assessment in Infants with Congenital or Early Onset Sensorineural Hearing Loss: Is Neonatal Vestibular Screening Required? A Scoping Review"

_audiolres, 2025, doi:10.3390/audiolres15020023_

Round 1
Reviewer 1 Report
Comments and Suggestions for Authors
The authors have done a scoping review of 14 articles with the objective to identify the prevalence of vestibular dysfunction in infants with congenital or early onset sensorineural hearing loss. The authors also have tried to identify which vestibular assessment tests are commonly used on this population and their sensitivity and specificity. The authors conclude that implementation of vestibular screening is warranted in infants with sensorineural hearing loss. Overall, the review is well-planned and systematically executed.
The specific comments are provided below:
Abstract
Line 16. Can define what is “early onset”
Introduction
There are many small paragraphs of two sentences which can be combined. For example, lines 44-48 and lines 55-58 can be combined.
Lines 38-39. What is the age range of the population studied in these studies?
Line 78-81. Needs reference
Line 84. Why is it specifically mentioned for “Ireland”?
Materials and Methods
Line 150. An updated search can be conducted to check if any additional articles are available.
Line 153. What does “Categorically logged” mean?
Line 157. Any software was used?
Line 178. Expansion of CASP needed
Conclusions
This section is too long and can be condensed
Appendix C – as a supplementary material, the ratings provided for each question for each study can be provided.
Comments on the Quality of English LanguageNil
Author Response
Many thanks for your feedback provided, it is very useful and we appreciate you taking the time to provide it. Please see responses to your feedback below.
Comments 1: Abstract Line 16. Can define what is “early onset”
Response 1: Added brackets (<12 months) to define what 'early onset' is in respect to our review.
Comments 2: Introduction There are many small paragraphs of two sentences which can be combined. For example, lines 44-48 and lines 55-58 can be combined.
Response 2: We have combined these as requested.
Comments 3: Introduction Lines 38-39. What is the age range of the population studied in these studies?
Response 3: We have now specified it is older paediatric populations studied.
Comments 4: Introduction Line 78-81. Needs reference
Response 4: Reference provided has been moved to the end of the sentence as information provided in this sentence was taken from this one reference.
Comments 5: Introduction Line 84. Why is it specifically mentioned for “Ireland”?
Response 5: Ireland was originally specifically mentioned as we hope that in the future, a vestibular assessment pathway could be implemented into our UNHS programme. In the context of this review, we have now removed the word 'Ireland'.
Comments 6: Materials and Methods Line 150. An updated search can be conducted to check if any additional articles are available.
Response 6: An updated search was conducted (February 2025) and 4 more articles were included in the review. This has been detailed in the methods section.
Comments 7: Line 153. What does “Categorically logged” mean?
Response 7: Line 155: Amended to remove the word 'categorically'.
Comments 8: Line 157. Any software was used?
Response 8: Line 155: Added 'Microsoft Excel' as this was used to log search results.
Comments 9: Line 178. Expansion of CASP needed
Response 9: Line 178: Expanded this to give more clarification as to why JBI Critical Appraisal tools were used.
Comments 10: Conclusions This section is too long and can be condensed
Response 10: The conclusion was condensed and some of the surplus material was added to the discussion.
Comments 11: Appendix C – as a supplementary material, the ratings provided for each question for each study can be provided.
Response 11: Completed JBI critical appraisal tools were added to Appendix C for greater transparency of ratings.
Reviewer 2 Report
Comments and Suggestions for Authors
Dear Authors, thank you for your manuscript. It is overall well-written, and I have a few suggestions.
1) Your stance on whether neonatal vestibular screening is required needs to be clearer. There is a clear lack of evidence at this moment for wide-scale adoption of cVEMP, for all infants born with congenital SNHL. I hear that prospective studies are needed, larger scale studies and standardized protocols of assessments are needed but it felt like a "food for thought" conclusion on whether you feel that at this moment, we can adopt universal cVEMP vestibular screening for example (likely no without more clinical evidence and also cost-effective analysis).
2) You have mentioned the limitations of the scoped studies, which is not limited to the heterogeneity in data collected (especially cVEMP). Different protocols of recording as we know will affect results and interpretation. This will directly affect prevalence estimation. Theoretically, we all agree in principle that infants with congenital SNHL may also have vestibular dysfunction due to the close proximity and embryological similarities. The most feasible and common test for young infants is agreed to be cVEMP as well. Since cVEMPS only assess saccular function, would an isolated saccular dysfunction (especially unilateral) truly affect functional motor development? In the older population, we know that cVEMPs are highly specific but poorly sensitive. The wide range of sensitivity and specificity in younger individuals is likely due to the heterogenity of the VEMP recordings (unstandardized protocols and normative data variations). Hence, prevalence is truly hard to estimate at this moment.
3) Are there insights to congenital disorders that you know that specifically targets the cochlear and saccule? Perhaps like in Pendred syndrome where there is also likely Mondini and large vestibular aqueduct. I know this is a scoping review and not candidacy for vestibular screening. However, if the intention is to encourage vestibular screening, I feel that the authors should make clearer suggestions on next step. Ie 1) Standardize protocol for prospective data collection on normal and abnormal VEMPS. 2) narrower and clear candidacy criteria on inclusion for most common used cVEMP screening for young infants-with correlate to? specific congenital inner ear disorders known to affect both hearing and balance. 3) Longitudinal follow up to observe and look at those with and without disease/vestibular dysfunction. Perhaps narrowing the focus to just VEMPS alone first (since it's probably going to be the 1st test used) will have greater clinical value.
4) A short comparison on the different Vemp protocols (stimulus level/head turn seated or neck flexion etc) used in the studies that reported SN/SP in your review, will be helpful for future studies to understand the differences in results. ? insufficient stimulation level ?In sufficient contraction of the SCM, BC versus AC (middle ear problems)?
Thank you for your work once again. It is nevertheless an important review.
Author Response
Thank you very much for your feedback, it has been very useful and we appreciate you taking the time to provide it. Please see responses to your feedback below.
Comments 1: Your stance on whether neonatal vestibular screening is required needs to be clearer. There is a clear lack of evidence at this moment for wide-scale adoption of cVEMP, for all infants born with congenital SNHL. I hear that prospective studies are needed, larger scale studies and standardized protocols of assessments are needed but it felt like a "food for thought" conclusion on whether you feel that at this moment, we can adopt universal cVEMP vestibular screening for example (likely no without more clinical evidence and also cost-effective analysis).
Response 1: We have made our stance clearer in the discussion section and added a separate section on 'Directions for Future Research' Lines 382-394. We also condensed the conclusion section to clarify our stance: Lines 412-420.
Comments 2: You have mentioned the limitations of the scoped studies, which is not limited to the heterogeneity in data collected (especially cVEMP). Different protocols of recording as we know will affect results and interpretation. This will directly affect prevalence estimation. Theoretically, we all agree in principle that infants with congenital SNHL may also have vestibular dysfunction due to the close proximity and embryological similarities. The most feasible and common test for young infants is agreed to be cVEMP as well. Since cVEMPS only assess saccular function, would an isolated saccular dysfunction (especially unilateral) truly affect functional motor development? In the older population, we know that cVEMPs are highly specific but poorly sensitive. The wide range of sensitivity and specificity in younger individuals is likely due to the heterogenity of the VEMP recordings (unstandardized protocols and normative data variations). Hence, prevalence is truly hard to estimate at this moment.
Response 2: We have taken your feedback into account and have expanded the limitations section (Lines 396-409).
Comments 3: Are there insights to congenital disorders that you know that specifically targets the cochlear and saccule? Perhaps like in Pendred syndrome where there is also likely Mondini and large vestibular aqueduct. I know this is a scoping review and not candidacy for vestibular screening. However, if the intention is to encourage vestibular screening, I feel that the authors should make clearer suggestions on next step. Ie 1) Standardize protocol for prospective data collection on normal and abnormal VEMPS. 2) narrower and clear candidacy criteria on inclusion for most common used cVEMP screening for young infants-with correlate to? specific congenital inner ear disorders known to affect both hearing and balance. 3) Longitudinal follow up to observe and look at those with and without disease/vestibular dysfunction. Perhaps narrowing the focus to just VEMPS alone first (since it's probably going to be the 1st test used) will have greater clinical value.
Response 3: Our section on 'Directions for Future Research' Lines 382-394 now addresses this piece of feedback.
Comments 4: A short comparison on the different Vemp protocols (stimulus level/head turn seated or neck flexion etc) used in the studies that reported SN/SP in your review, will be helpful for future studies to understand the differences in results. ? insufficient stimulation level ?In sufficient contraction of the SCM, BC versus AC (middle ear problems)?
Response 4: We have now added an additional (fifth) extraction table to address this piece of feedback. The fifth extraction table can be found from Lines 262.
Reviewer 3 Report
Comments and Suggestions for Authors
Dear authors,
Thank you for this interesting manuscript.
Please find my recommendations here in the following lines:
General:
- Did you analyze the rate of successful measurements: Was a measurement possible at all without paying attention to correct response or no response? As infants are hard to test in audiometry I expect also difficulties in neurotolgy.
- Extensively discussion the differences of performing these tests in infants / children instead of adults. Some tests can also be challening in adults. Therefore, testing children may be impossible sometimes.
14: ... in infant ...
18ff: missing verb
25: utilized within --> for
53: remove "itself"
61: replace "impact"
69: Please clarify this sentence
82ff: Please re-write this sentence
104: Prior to --> Before & was -> were
109: remove "the purpose of"
111: required both --> change wording
116: which --> that (same for some following sentences)
123: remove "time" & "the"
131: remove the comma
139ff: Google Scholar
141: using
145: remove "in order"
150: remove "the"
155: additional
157ff: were
159: ..., in case ...
161: "Full-text" & rewrite "articles deemed potentially eligible"
168: prior to --> before & maybe something like that fits better "The review objectives were analyzed before designing appropriate extraction tables that chart all relevant data."
181: objectives & replace "relating to" by "on"
182: in regard to --> regarding
194: re-write "in a manner that reflects"
Fig. 1: The image quality is very low and needs improvement. I can only read the text when using the zoom of the pdf reader. And Phase2->3: Why is 163 minus 100 = 54? There seems to be a typo. In addition, the last step "Bibliographies of ..." should be visualized the same way as the exlusions before.
203: remove "of"
204: follow-up
205ff: use active voice
206: remove "ranging"
207: remove "and"
207f: Most articles ...
210f: re-write this sentence
212: replace "consisting of"
214: was assessed
222ff table:
- Please add "Total: " in all cells of "Number of Subjects (n)" if there is more information like x cases and y controls.
- "Verrecchia et al., 2019": "2016 (14-month period)" should be corrected to 2015-2016, 2016-2017, or 2015-2017 due to the period > 1y.
227: remove comma
231: using
232: remove "used"
234: remove ", with degrees"
238 table:
- please always provide the manufacturer and the name of the hardware. Currently we have sometimes in brakets MAICO or ALGO or nothing --> MAICO beraphone / natus ALGO / unkown / ...
- Text orientation is not always the same in this table see "Martens et al., 2022*" or "Unilateral: Mild-moderate (11.8%)," --> please fix and also control all tables in this manuscript
- "Click-evoked ABR, high frequency tympanometry (1000Hz), T/DOAEs." -->
- high-frequency
- TE/DPOAEs
- c-ABR = Click-ABR?
- "Normal tympanogram, referred DPOAE (<4 points passed in the 6 selected frequencies), elevated air c-ABR threshold (>30dBnHL), air and bone-conducted c-ABR threshold gap within 10dBnHL. 2,000 and 4,000 Hz TB-ABR, Auditory Steady-State Response (ASSR)" --> needs to be shorter using, e.g., 1 kHz for 1000Hz and ASSR can be defined under the table.
247: re-write "the use of", e.g., "Another article reported caloric measurements and cVEMPs in combination on infants aged three months."
262 table: "Maes et al., 2017": Bullet point missing before "Unilateral SNHL (3): 1/3 unilateral VD, 1/3 bilateral VD, 1/3 present response."
269: remove 2nd "to"
279: remove comma
281: Explain how a rotational chair works with children regarding fixation of the infant / child on the chair. This sound's very challenging to me.
291: child-friendly
293: was -> were
314: missing verb
317: vHIT of ...
320: a sensitifity ...
321: the validity ...
323: ... were
Make "Limitations" a section and extend its content regarding all kind of possible biases, number of articles and number of patients in articles availible in journals, reliabilty of the data (signal to noise ratio when testing infants / children), ...
334: Although this scoping review provides ...
336: "several" instead of "a number of"
345f: please re-write this sentence to improve its clarity
352: re-write "provided a sound presentation"
353: on vestibular ...
354: signal without the "s" & remove "for"
357: over-burdened
359: remove "the" & "Further targeted assessments should also be conducted on infants with specific risk factors for comorbidity like but not limited to meningitis, cCMV, rubella, and those exposed to ototoxic medications."
360: remove "the"
362: fulfills
363: management of ...
364: remove "with"
370ff: Clinicians can acquire more diagnostic data by assessing the vestibular system, which may help identify the underlying cause of the hearing loss.
372ff: re-write this sentence
377: However, ...
379f: bone-conducted
380f: such as -> like
382: remove "with reports"
383f: re-write this sentence to improve clarity
384: remove comma
386: remove "on"
386ff: The few articles available to review the research question demonstrate the overall lack of meaningful evidence in support of curating national policy changes or intervention implementation.
390ff: re-write this sentence to improve clarity
392: However, ...
393: cut-off & "substantiated "a priori" to"
394: re-write "to better inform the research question" & "Overall, this scoping review illustrates promising results both in feasibility and diagnostic accuracy of VD in the target population."
396: literature without the "s"
396ff: re-write this sentence to improve clarity and split it into a minimum of to sentences.
410: aABR --> AABR (aABR is also used for acoustic ABR in contrast to eABR for electrical stimulation)
Comments on the Quality of English Language
- Please use US english all over the text, ...
- ... use "that" instead of "which", ...
- ... and use "..., ..., and ..." without missing the last comma.
Author Response
Thank you so much for your feedback, it is very useful and we appreciate you taking the time to provide it. Please see our responses below to your feedback.
Comments 1: Did you analyze the rate of successful measurements: Was a measurement possible at all without paying attention to correct response or no response? As infants are hard to test in audiometry I expect also difficulties in neurotolgy.
Response 1: We have addressed this piece of feedback from Lines 262-268.
Comments 2: Extensively discussion the differences of performing these tests in infants / children instead of adults. Some tests can also be challening in adults. Therefore, testing children may be impossible sometimes.
Response 2: We have addressed this piece of feedback from Lines 368-376.
Comments 3: 14: ... in infant ...
18ff: missing verb
25: utilized within --> for
53: remove "itself"
61: replace "impact"
69: Please clarify this sentence
82ff: Please re-write this sentence
104: Prior to --> Before & was -> were
109: remove "the purpose of"
111: required both --> change wording
116: which --> that (same for some following sentences)
123: remove "time" & "the"
131: remove the comma
139ff: Google Scholar
141: using
145: remove "in order"
150: remove "the"
155: additional
157ff: were
159: ..., in case ...
161: "Full-text" & rewrite "articles deemed potentially eligible"
168: prior to --> before & maybe something like that fits better "The review objectives were analyzed before designing appropriate extraction tables that chart all relevant data."
181: objectives & replace "relating to" by "on"
182: in regard to --> regarding
194: re-write "in a manner that reflects"
Responses 3: We have amended these revisions in the main text as requested.
Comments 4: Fig. 1: The image quality is very low and needs improvement. I can only read the text when using the zoom of the pdf reader. And Phase2->3: Why is 163 minus 100 = 54? There seems to be a typo. In addition, the last step "Bibliographies of ..." should be visualized the same way as the exlusions before.
Response 4: A clearer PRISMA flowchart was created which hopefully rectifies this issue of it not being easy to read. Typo has been corrected. An updated search was conducted in February 2025 to include any additional articles in the review and this is noted on the PRISMA flowchart.
Comments 5: 203: remove "of"
204: follow-up
205ff: use active voice
206: remove "ranging"
207: remove "and"
207f: Most articles ...
210f: re-write this sentence
212: replace "consisting of"
214: was assessed
222ff table:
- Please add "Total: " in all cells of "Number of Subjects (n)" if there is more information like x cases and y controls.
- "Verrecchia et al., 2019": "2016 (14-month period)" should be corrected to 2015-2016, 2016-2017, or 2015-2017 due to the period > 1y.
227: remove comma
231: using
232: remove "used"
234: remove ", with degrees"
Response 5: We have amended these revisions in the main text as requested.
Comments 6: 238 table:
- please always provide the manufacturer and the name of the hardware. Currently we have sometimes in brakets MAICO or ALGO or nothing --> MAICO beraphone / natus ALGO / unkown / ...
- Text orientation is not always the same in this table see "Martens et al., 2022*" or "Unilateral: Mild-moderate (11.8%)," --> please fix and also control all tables in this manuscript
- "Click-evoked ABR, high frequency tympanometry (1000Hz), T/DOAEs." -->
- high-frequency
- TE/DPOAEs
Response 6: Equipment manufacturer has been noted in this table, if the manufacturer was not reported in the article, we have stated that it is unknown. We have amended text orientation and controlled tables included. We have amended added hyphenated 'high-frequency' and amended 'T/DOAEs' to 'TE/DPOAEs'.
Comments 7: c-ABR = Click-ABR?
Response 7: We have now amended c-ABR to 'Click-ABR' to improve clarity.
Comments 8: "Normal tympanogram, referred DPOAE (<4 points passed in the 6 selected frequencies), elevated air c-ABR threshold (>30dBnHL), air and bone-conducted c-ABR threshold gap within 10dBnHL. 2,000 and 4,000 Hz TB-ABR, Auditory Steady-State Response (ASSR)" --> needs to be shorter using, e.g., 1 kHz for 1000Hz and ASSR can be defined under the table.
Response 8: This has been shortened with definitions provided at the end of the table.
Comments 9: 247: re-write "the use of", e.g., "Another article reported caloric measurements and cVEMPs in combination on infants aged three months."
262 table: "Maes et al., 2017": Bullet point missing before "Unilateral SNHL (3): 1/3 unilateral VD, 1/3 bilateral VD, 1/3 present response."
269: remove 2nd "to"
279: remove comma
Response 9: We have amended these revisions in the main text as requested.
Comments 10: 281: Explain how a rotational chair works with children regarding fixation of the infant / child on the chair. This sound's very challenging to me.
Response 10: The method of how rotational chair was conducted with study subjects in Lines 301-306 and also more information has been included in the relevant extraction table (260ff).
Comments 11: 291: child-friendly
293: was -> were
314: missing verb
317: vHIT of ...
320: a sensitifity ...
321: the validity ...
323: ... were
Response 11: We have amended these revisions in the main text as requested.
Comments 12: Make "Limitations" a section and extend its content regarding all kind of possible biases, number of articles and number of patients in articles availible in journals, reliabilty of the data (signal to noise ratio when testing infants / children), ...
Response 12: We have made the 'limitations' a separate section and have extended it to include more detailed limitations (395ff)
Comments 13:
334: Although this scoping review provides ...
336: "several" instead of "a number of"
345f: please re-write this sentence to improve its clarity
352: re-write "provided a sound presentation"
353: on vestibular ...
354: signal without the "s" & remove "for"
357: over-burdened
359: remove "the" & "Further targeted assessments should also be conducted on infants with specific risk factors for comorbidity like but not limited to meningitis, cCMV, rubella, and those exposed to ototoxic medications."
360: remove "the"
362: fulfills
363: management of ...
364: remove "with"
370ff: Clinicians can acquire more diagnostic data by assessing the vestibular system, which may help identify the underlying cause of the hearing loss.
372ff: re-write this sentence
377: However, ...
379f: bone-conducted
380f: such as -> like
382: remove "with reports"
383f: re-write this sentence to improve clarity
384: remove comma
386: remove "on"
386ff: The few articles available to review the research question demonstrate the overall lack of meaningful evidence in support of curating national policy changes or intervention implementation.
390ff: re-write this sentence to improve clarity
392: However, ...
393: cut-off & "substantiated "a priori" to"
394: re-write "to better inform the research question" & "Overall, this scoping review illustrates promising results both in feasibility and diagnostic accuracy of VD in the target population."
396: literature without the "s"
396ff: re-write this sentence to improve clarity and split it into a minimum of to sentences.
Response 13: We have amended these revisions in the main text as requested.
Comments 14: 410: aABR --> AABR (aABR is also used for acoustic ABR in contrast to eABR for electrical stimulation)
Response 14: We have amended aABR to AABR to improve clarity.
Comments 15: Please use US english all over the text, ...
Response 15: We have amended to use US english throughout
Comments 16: ... use "that" instead of "which", ...
... and use "..., ..., and ..." without missing the last comma.
Response 16: We have amended this throughout.